# Factors Influencing Social Media Users' Continued Intent to Donate

**Yi Li and Liangru Yu ***

School of Economics and Management, Chongqing University of Posts and Telecommunications, Chongqing 400065, China; liyi@cqupt.edu.cn

*** Correspondence: yuliangru@hotmail.com

**Abstract:** Social media donation is an emerging sustainable business model. Donation in the context of social media can often bring benefits to content creators and social media platforms, as well as realizing their sustainable development. Based on attachment theory, customer loyalty theory, and interaction ritual chains theory, this paper studies how feedback interaction and participatory interaction affect users' continued intent to donate. The role of users' emotion and price consciousness are mainly considered. Data were collected through questionnaires, and the sample covered 466 WeChat users. Structural equation modeling and linear regression were used to test the hypothesis. It was found that emotional attachment and emotional loyalty had significant positive effects on users' continued intent to donate, and participatory interaction had significant positive effects on emotional attachment and emotional loyalty, while feedback interaction had a significant positive effect on emotional attachment. Price consciousness did not directly affect continued intent to donate, but as a moderator, it weakened the positive relationship between emotional attachment and continued intent to donate.

**Keywords:** continued intent to donate; interaction; emotional attachment; emotional loyalty; price consciousness; structural equation modelling

## 1. Introduction

Social media are highly interactive platforms based on mobile and network technology, on which individuals and communities share, create, discuss and modify user-generated content [1]. The more content that creators generate on social media, the more influential the platforms will be. Therefore, to motivate content creators to generate content, social media platforms have introduced a new function: donation. Donation on social media means that users donate to content creators based on the knowledge and services they provide, and content creators can receive monetary or nonmonetary rewards as a result [2].

On social media, donation is an income source for content creators [3]. To ensure the innovation and long-term survival of social media platforms, platforms share donations with content creators who receive donations [4]. Therefore, in order for content creators and social media platforms to be consistently profitable, users need to be encouraged to continue to donate. As a result, users' continued intent to donate should be studied. For example, WeChat, one of the largest social media platforms in China, had more than 10 million public accounts (which provide content for users to read and consequently receive donations) by the end of 2017. Users do not need to pay to read articles, and donation is discretionary behavior, resulting in a low proportion of donations. Some users do not donate, some occasionally donate, and some donate continually. Whether the business model of donation is sustainable depends on continued donation from users. Therefore, there is an urgent

need to study users' continued intent to donate on social media platforms. However, there are few relevant studies.

Based on a review of the existing literature, Liu [5] believed that perceived value was a key factor in determining donation behavior, and Wan et al. [2] thought that donation intention was determined by emotional attachment to the content creator and functional dependence on social media. There have been articles on user donation in the context of social media, but the factors that affect continued donation have not been fully discussed. This paper considers how users interact with content creators and other users on social media. Interaction is the main reason why social media are different from other media and can keep platforms alive because users form relationships and maintain social networks in the process of interacting [6]. Therefore, we chose interaction ritual chains and divided interaction rituals into feedback and participatory interaction according to the social media context [7]. Successful interaction rituals will result in the gain of a certain amount of emotional energy, which makes consumers more enthusiastic about interaction rituals; the energy can even influence subsequent decisions [8]. Therefore, this paper considers users' emotional factors, looking at two kinds of long-term continuous emotions—emotional attachment and emotional loyalty—and combines attachment theory and customer loyalty theory to research users' continued intent to donate. In addition, personal emotions are susceptible to other factors. Considering that some personal traits can moderate the behavioral intention generated by emotion, we chose price consciousness as a moderator. In summary, this paper studies users' continued intent to donate in the context of social media. It introduces interaction rituals (feedback interaction, participatory interaction), emotional energy (emotional attachment, emotional loyalty), and price consciousness factors. Based on interaction ritual chains, attachment theory, and customer loyalty theory, this paper constructed a model of social media users' continued intent to donate, which is driven by interaction rituals and emotional energy (see Figure 1).

The contribution of this paper differs from existing research in the following ways. First, we studied users' continued intent to donate in the context of social media. There have been studies on donation in the context of social media, but they did not focus on continued donation. Second, we propose a new theoretical framework by integrating attachment theory, customer loyalty theory, and interaction ritual chains. Within the context of social media, we divide interaction rituals into feedback and participatory interaction, and explore the differences between the two. We study the continued intent to donate from the perspective of interaction between users and content creators; that is, we study the antechambers of users' continued intent to donate, using the framework of interaction rituals–emotional energy–behavioral intention. Third, we select price consciousness as the moderator according to behavioral characteristics of social media users.

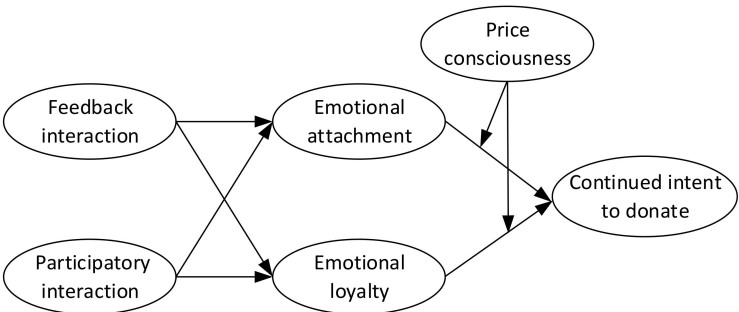

**Figure 1.** Model of users' continued intent to donate.

## 2. Literature Review

### 2.1. Donation on Social Media

Donation first appeared in the context of busking, in which the audience gave a certain amount of money according to their intentions after the entertainer's performance [9]. With the development of

the Internet and the rise of online payment, donation has emerged on social media platforms; that is, users donate to others with virtual gold coins, virtual gifts, or cash on social media platforms [2]. In recent years, the donation function has become widely popular. For example, Sina Weibo (the most famous microblogging platform in China) launched a donation function in 2014 and obtained about 7 million dollars in revenue through donation in 2015 [5]. The donation function is a new business model for content payment on social media platforms, which combines donation and user-generated content (UGC) to further realize the commercialization of original content.

Existing literature shows that there are three main factors influencing donation from social media users': users, content creators and social media platforms. Specifically, social media users will make an overall assessment of the benefits and sacrifice of the content they read (perceived value), which is the basis for determining donation behavior [5]. Some users use support for content creators (donation behavior) as a way to seek more social interaction with other people. For example, in live video streaming, audience participation is positively correlated with gift-giving behavior [10]. In addition, emotional attachment to content creators [2,11], their reputation and two trustworthiness attributes (ability and integrity) [12] have a positive impact on payment/donation decisions of social media users. In the context of social media, some users will show their love and appreciation for the personal charm of content creators. For users, donation has become an effective way to convey deep feelings to content creators. Finally, the functional dependence of users on social media can affect their willingness to donate [2]. If a user is functionally dependent on social media, he or she will spend more time or money on the social media platform. This kind of social media functional dependence helps to enhance users' intention to donate.

## 2.2. Attachment Theory and Social Media Users' Sustained Behavior

Many studies on early attachment theory were conducted by John Bowlby and Mary Ainsworth [13], focusing on attachment in children and the emotional bonds formed by individuals with objects during their growth [14]. Later, attachment theory was studied in romantic love [15], family members, and friends [16]. Now, attachment research has moved into various fields. Existing studies have shown that in the context of relationship marketing, attachment is used to explain the relationship between consumers and brands [17], and brand attachment is considered to reflect the strength of the cognitive and emotional bond connecting people with brands [18].

Ren et al. [19] divided attachment in online communities into common identity attachment and common bond attachment, which influence users' behaviors in virtual communities [20]. Therefore, attachment theory needs to consider two aspects: on the one hand, members identify with the whole group (based on identity attachment), and on the other hand, members like individuals in the group (based on bond attachment). Similarly, social media are communities, so social media users may pay more attention to their common interests (common identity attachment) or establish relationships with others in the community (common bond attachment) [21]. The process of forming these two attachments leads to closer relationships between social media users and content creators or other users. In other words, social media users may form emotional attachments through this process.

Consumers with an emotional attachment are willing to put resources such as time, energy, and money into target objects [18,22] to maintain their relationship with those objects [23]. To sum up, emotional attachment can create a strong incentive for consumers to devote personal resources to maintain relationships, manifested by emotionally ingrained repurchasing and avoidance of conversion [24]. So, in the context of social media, we can use attachment theory to explain why users continue to donate to content creators.

## 2.3. Customer Loyalty Theory and Social Media Users' Sustained Behavior

Reviewing the existing literature, customer loyalty can often be explained by attitudes and behaviors [25]. Chaudhuri and Holbrook [26] suggested that behavioral or purchasing loyalty consists of the repeated purchase of a brand, while attitude loyalty consists of a certain degree of character

commitment, that is, some unique value associated with the brand. Other scholars believe that attitude loyalty also includes consumers' preferences and intentions [27,28].

Some scholars have proposed that emotional loyalty is one of the modes of customer loyalty; it shows a deep emotional connection with the brand. Emotionally loyal users do not use other products or services [29]. To sum up, in the context of social media, users who are emotionally loyal to a content creator will have a deep emotional connection with the content creator and pay less attention to content posted by others. This type of user shows continued donation behavior. Therefore, in the context of social media, we can use customer loyalty theory to explain why users continue to donate to content creators.

### 2.4. Social Media Use of Interaction Ritual Chains

Collins pointed out that the formation of interaction ritual chains included four aspects: (1) there were two or more people gathered in the same place; (2) there were clear boundaries between inside and outside; (3) they had a common focus; and (4) people shared common emotions or emotional experiences. Collins [8] believed that at the beginning of the interaction ritual, two or more individuals gathered together because of their common focus of attention. At this time, individuals only invested in temporary emotions. After experiencing long-term interactive rituals conversion, the temporary emotions were then transformed into long-term emotional energy [8]. Therefore, we can find that emotional energy is a lasting emotion generated after the long-term transformation of interactive rituals and can affect people's decision-making.

Scholars have applied interaction ritual chains to online networks [30]. In the context of social media, the presence of virtual space breaks through the limitation of a common presence in physical space. Users have common concerns about content creators, and through internal sharing of information content, emotional communication and interaction can be realized so that interaction ritual chains can be formed. Therefore, this paper uses interaction ritual chains to explain the emotional energy generated by interaction rituals in the context of social media.

## 3. Hypothesis

### 3.1. Emotional Energy

Emotional attachment is a relationship-based concept that reflects the emotional connection between individuals and consumption entities (such as brands, people, places, or objects) [31]. Previous studies considered that consumers who were attached to brands were willing to invest their resources in the brands to maintain relationships with them [2,22,31]. In the context of social media, content creators are spokespersons for content about products, and they have a brand effect. Continued donation means that users continue to invest resources (money) into content creators, and users with an emotional attachment to content creators are likely to continue donating to them. Therefore, we propose the following hypothesis:

**Hypothesis 1.** *Emotional attachment is positively related to users' continued intent to donate.*

Emotional loyalty refers to the degree of positive emotion caused by repurchasing a brand [32]. Some scholars have proposed that emotional loyalty is one of the modes of customer loyalty [29]. Oliver [33] believed that loyalty is the highest level of commitment, which meant that the change from a favorable tendency (emotional loyalty) to a repeated purchase commitment (conative loyalty) was the first step in repeated purchase (behavioral loyalty). To sum up, in the context of social media, users who have emotional loyalty to content creators will not be influenced by the external environment or other brand marketing strategies and will generate preference and repeated purchase commitment to the content creators. Continued donation can be viewed as repurchase, meaning behavioral loyalty to the

content creator. So, when users are emotionally loyal to a content creator, they will continue to donate to the content creator's articles. Therefore, we propose the following hypothesis:

**Hypothesis 2.** *Emotional loyalty is positively related to users' continued intent to donate.*

*3.2. Interaction Rituals*

Muirhead [7] defined interactivity as "communication, participation and feedback." Bonner [34] divided interaction into three dimensions of two-way communication, participation, and joint problem-solving in a business-to-business (B2B) environment. Therefore, according to the characteristics of social media, interaction rituals are divided into two categories: feedback and participatory interaction. Feedback interaction refers to communication between users and content creators, which is reflected in the replies, feedback, and interactions between the two sides. Participative interaction refers to unilateral participative behavior by users toward content creators, which is represented by reposts, collects, likes, and so on.

Altman and Taylor Hudson [35] believed that frequent social interactions between two people led to greater interdependence and intimacy. In the context of social media, interactions with brands could cultivate intimacy as well as interpersonal social interaction [36]. Feedback interaction is the interpersonal social interaction between users and content creators, and emotional attachment is the intimate emotion between people. So we believe that users will have an emotional attachment to content creators through feedback interaction. Therefore, we propose the following hypothesis:

**Hypothesis 3.** *Feedback interaction is positively related to users' emotional attachment.*

Social media are virtual communities. Through participatory interaction, users generate an emotional sense of connection and feel that they belong to the community, making it is easy to produce attachment to the community. Just as social participation in a community fosters attachment to the community [37], so we think that users will have an emotional attachment to content creators through participatory interaction. Therefore, we propose the following hypothesis:

**Hypothesis 4.** *Participatory interaction is positively related to users' emotional attachment.*

In the context of social media, when users begin to interact with content creators (feedback interaction), they have a certain relationship with the content creators and will try to confirm their close relationship with content creators through more frequent interactions (feedback interactions), so as to form emotional loyalty to the content creators. Dholakia and Durham [38] found that on social media, interaction levels influenced behavioral loyalty. Therefore, we propose the following hypothesis:

**Hypothesis 5.** *Feedback interaction is positively related to users' emotional loyalty.*

In online networks, customers participating in a brand's marketing process generate website brand loyalty [39]. In the context of social media, content creators are the spokesmen of brands, and they generate content to promote themselves. So, we think that in the context of social media, users participate in content generated by content creators through interaction (participatory interaction) and then generate emotional loyalty. Therefore, we propose the following hypothesis:

**Hypothesis 6.** *Participatory interaction is positively related to users' emotional loyalty.*

*3.3. The Moderating Effect of Users' Personal Traits*

In online purchasing, consumer traits are related to intention, adoption, and persistence [40]. Social media donation is similar to online purchase, so we selected the user characteristics of price consciousness for research. Price consciousness refers to the speed and extent of consumers'

psychological response to changes or differences in the price of products or services [41]. Previous studies used price consciousness as a moderator for continuous purchase. Graciola [42] believed that store price impressions would affect repurchase intentions, and the strength of the relationship depended on the moderating effect of price consciousness. This study suggests that in the context of social media, people who are not conscious of price are more willing to pay than people who are. This is because the former are more cautious about price, which will eliminate or reduce behavior driven by emotion (emotional attachment, emotional loyalty), so they will make more rational decisions regarding their continued intent to donate. Therefore, we propose the following hypothesis:

**Hypothesis 7.** *Price consciousness plays a negative moderating role between emotional attachment and continued intent to donate.*

**Hypothesis 8.** *Price consciousness plays a negative moderating role between emotional loyalty and continued intent to donate.*

## 4. Methodology

In 2018, with more than 1 billion users in China, WeChat became one of the most important and popular mobile social media platforms in China. The WeChat public account is currently the most influential content platform. Compared with other social media platforms, it has the highest percentage of user donations, so WeChat donation is highly representative of social media donation. Its donation function allows users to voluntarily pay a fee of their choosing after reading content generated by content creators. In this study, we took WeChat users as the research object and collected data through an online questionnaire survey to test our proposed model.

### 4.1. Participants and Process

In this study, we collaborated with a professional online survey company (paid) to sample, design questionnaires online, generate web links, and then distribute them randomly in a 2.6 million sample bank. A total of 2264 questionnaires were collected from 4014 people randomly selected by a computer program, for a recovery rate of 56.40%. A total of 466 valid questionnaires (187 from men and 279 from women) were obtained. Referring to other social media donation studies [2], 466 valid questionnaires is an acceptable number. We strictly excluded questionnaires with abnormal information or logical errors, careless filling in, or mechanical answers. We asked the participants to write the name of the public account and the content creator which opened the donation function, and then we checked whether the information was correct. Questionnaires in which one of the two pieces of information was incorrect were considered invalid. To find out whether there is a correlation between sex and intention to donate, we conducted an independent sample t-test on six variables (feedback interaction, participatory interaction, emotional attachment, emotional loyalty, price consciousness, continued intent to donate). The results showed that there was no significant difference in the mean value of the six variables between the men and women ($p > 0.05$). The respondents were all under 50 years of age (M = 27.12, SD = 6.20). In terms of monthly income, 3.43% earned less than 1000 CNY, 13.52% earned between 1000 and 2000 CNY, 4.51% earned between 2000 and 3000 CNY, 5.58% earned between 3000 and 4000 CNY, 11.16% earned between 4000 and 5000 CNY, 11.37% earned between 5000 and 6000 CNY, 11.37% earned between 6000 and 7000 CNY, 10.73% earned between 7000 and 8000 CNY, 6.65% earned between 8000 and 9000 CNY, 9.01% earned between 9000 and 10,000 CNY, and 12.66% earned more than 10,000 CNY.

### 4.2. Measurement

This study involved 6 constructs: feedback interaction, participatory interaction, emotional attachment, emotional loyalty, price consciousness, and continued intent to donate. All constructs were measured by the Likert scale: 1 represented very inconsistent and 7 represented very consistent.

The scale of feedback interaction was defined by Kuo and Feng [43]. The scale of participatory interaction was designed according to the function of WeChat. The scale of emotional attachment was defined by Ren et al. [44]. The scale of emotional loyalty was defined by Lam et al. [45]. The scale of price consciousness was defined by Kim et al. [46]. The scale of continued intent to donate was defined by Bhattacherjee [47]. In this paper, age and income were introduced as control variables [48], and age and income were logarithmically processed.

Since all respondents were Chinese, we translated the scale from English to Chinese and made appropriate modifications according to the use scenarios of WeChat. For example, the original version of the attachment scale was, "I would like to be a friend of [member name]," and it was changed to, "I would like to be a friend of this author." The measurement items of each variable are shown in Table 1.

**Table 1.** Measurement items for each variable.

| Construct | Items | References |
|---|---|---|
| Continued intent to donate | I intend to continue donating to the author. | Bhattacherjee [47] |
| | My intentions are to continue donating to the author rather than donating to any other authors. | |
| | If I could, I would like to continue donating to the author. | |
| Emotional attachment | I would like to be friends with the author. | Ren et al. [44] |
| | I would like to interact with the author. | |
| | I am interested in learning more about the author. | |
| Emotional loyalty | I would encourage friends to read the author's article. | Lam et al. [45] |
| | If someone ask me for my opinions, I will say positive things about the author. | |
| | I will recommend the author to those who seek my advice. | |
| Feedback interaction | My message can result in a quick reply from the author. | Kuo and Feng [43] |
| | I have close and intensive interactions with the author. | |
| | My comments and suggestions can result in quick feedback from the author. | |
| Participatory interaction | I often COLLECT the author's articles. | WeChat's function |
| | I often FORWARD the author's articles. | |
| | I often LIKE the author's article. | |
| Price consciousness | Before I buy a product, I often check the prices of different retailers to obtain the best benefit. | Kim et al. [46] |
| | I usually only purchase items on sale. | |
| | I usually purchase the cheapest item. | |

### 4.3. Reliability and Validity Test

SPSS 23 was used to test the reliability of the data. Cronbach's α values for feedback interaction, participatory interaction, emotional attachment, emotional loyalty, price consciousness, and continued intent to donate were 0.882, 0.728, 0.753, 0.726, 0.740, and 0.823, respectively. LISREL 8.8 was used to test the validity of the data, and the results of confirmatory factor analysis (CFA) showed that the 6-factor model fit well, based on the following statistics: chi-square ($\chi^2$) = 241.8, degree of freedom (df) = 120, $\chi^2$/df = 2.015, root mean square error of approximation (RMSEA) = 0.0467, non-normed fit index (NNFI) = 0.974, comparative fit index (CFI) = 0.980, incremental fit index (IFI) = 0.980, goodness of fit index (GFI) = 0.945, adjusted goodness of fit index (AGFI) = 0.922. However, the substituted Haman single factor model fit poorly: $\chi^2$ = 1767.51, DF = 135, $\chi^2$/DF = 13.093, RMSEA = 0.161, NNFI

= 0.693, CFI = 0.729, IFI = 0.730, GFI = 0.703, AGFI = 0.624. Therefore, the common method bias is not serious.

## 4.4. Model Test

We used the full model of structural equation modeling to test hypothesis H1–H6. We used SPSS 23 to test the moderating effects of hypothesis H7 and H8.

We wanted to study the relationship between latent factors, so we conducted structural equation modeling [2,49]. Using LISREL 8.8 software, the full model analysis method was used to analyze sample data in order to obtain a standardized estimate of the path coefficient of the model and its hypothesis test results (see Figure 2 and Table 2). It also obtained the following goodness-of-fit statistics: chi-square ($\chi^2$) = 325.36, degrees of freedom (df) = 107, $\chi^2$/df = 3.04, root mean square error of approximation (RMSEA) = 0.0662, non-normed fit index (NNFI) = 0.954, comparative fit index (CFI) = 0.964, incremental fit index (IFI) = 0.964, goodness of fit index (GFI) = 0.924, and adjusted goodness of fit index (AGFI) = 0.891, indicating that the degree of fit of the model to the data is acceptable. The path analysis parameter estimation listed in Table 1 shows that hypothesis H1 ($\beta$ = 0.43, $p < 0.01$), H2 ($\beta$ = 0.34, $p < 0.01$), H3 ($\gamma$ = 0.28, $p < 0.01$), H4 ($\gamma$ = 0.46, $p < 0.01$), and H6 ($\gamma$ = 0.63, $p < 0.01$) are supported, while H5 ($\gamma$ = 0.08, $p > 0.05$) is not.

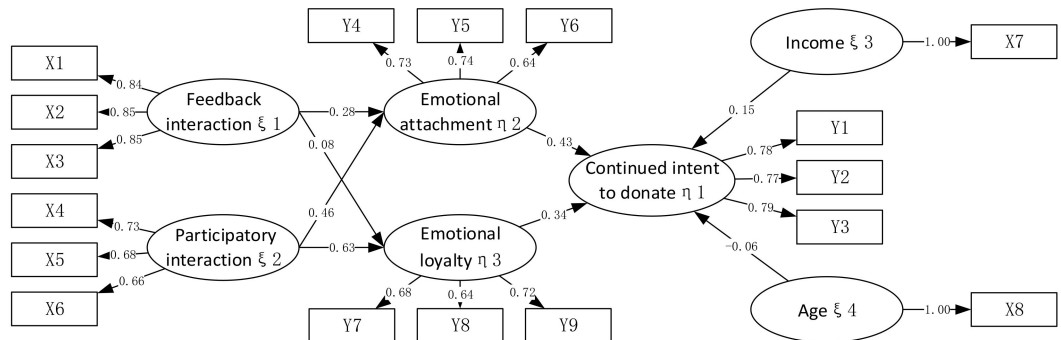

**Figure 2.** Completely standardized path coefficients for the full model. (X1, X2, . . . X8, Y1, Y2, . . . Y9 are measurement items; numbers on the path are standardized coefficients).

**Table 2.** Path analysis and hypothesis test results.

| Hypothesis | Paths | Standardized Estimate | *t*-Values | Supported |
|---|---|---|---|---|
| H1 | Emotional attachment (η2) → Continued intent to donate (η1) | 0.43 | 6.81 ** | Yes |
| H2 | Emotional loyalty (η3) → Continued intent to donate (η1) | 0.34 | 5.47 ** | Yes |
| H3 | Feedback interaction (ξ1) → Emotional attachment (η2) | 0.28 | 4.68 ** | Yes |
| H4 | Participatory interaction (ξ2) → Emotional attachment (η2) | 0.46 | 6.80 ** | Yes |
| H5 | Feedback interaction (ξ1) → Emotional loyalty (η3) | 0.08 | 1.40 | Not |
| H6 | Participatory interaction (ξ2) → Emotional loyalty (η3) | 0.63 | 8.04 ** | Yes |

Notes: **$p < 0.01$.

### 4.5. Test of Moderating Variables

We built four models to test these hypotheses (see Table 3). Model 1 examined the relationship between control variables and continued intent to donate. In model 2, we included emotional attachment and emotional loyalty. In model 3, we included a moderator (price consciousness). Finally, all variables and cross-terms were included in model 4. The variance inflation factor of the variables was less than 2 in all models.

According to the results of model 2 ($R^2$ = 0.275), emotional attachment ($\beta$ = 0.334, $p < 0.001$) and emotional loyalty ($\beta$ = 0.229, $p < 0.001$) had significant positive impact on continued intent to donate. According to the results of model 3 ($R^2$ = 0.276), the relationship between price consciousness and continued intent to donate was not significant ($\beta$ = −0.039, $p > 0.05$). This reflected that price consciousness had no direct impact on continued intent to donate. The results of model 4 ($R^2$ = 0.284) showed that the cross-term between emotional attachment and price consciousness had a significant negative impact on continued intent to donate ($\beta$ = −0.1, $p < 0.05$). Therefore, hypothesis 7 is supported. However, the cross-term between emotional loyalty and price consciousness had no significant effect on continued intent to donate ($\beta$ = 0.042, $p > 0.05$). Therefore, hypothesis 8 is not supported.

**Table 3.** The results of the moderating effect.

| Variable | Dependent Variable: Continued Intent to Donate | | | |
| --- | --- | --- | --- | --- |
| | **Model 1** | **Model 2** | **Model 3** | **Model 4** |
| Constant | | | | |
| Income | 0.252 *** | 0.181 *** | 0.172 ** | 0.161 ** |
| Age | −0.072 | −0.050 | −0.050 | −0.034 |
| Emotional Attachment | | 0.334 *** | 0.336 *** | 0.341 *** |
| Emotional Loyalty | | 0.229 *** | 0.227 *** | 0.233 *** |
| Price Consciousness | | | −0.039 | −0.029 |
| Emotional Attachment X Price Consciousness | | | | −0.100 * |
| Emotional Loyalty X Price Consciousness | | | | 0.042 |
| $R^2$ | 0.048 | 0.275 | 0.276 | 0.284 |
| F | 11.551 *** | 43.675 *** | 35.116 *** | 25.990 *** |
| Max VIF | 1.522 | 1.546 | 1.596 | 1.615 |

Notes: (1) coefficient had been standardized; (2) * $p < 0.05$, ** $p < 0.01$, *** $p < 0.001$; (3) $R^2$: coefficient of determination; (4) VIF: variance inflation factor.

## 5. Discussion

The purpose of this study is to examine the impact of emotional energy (emotional attachment, emotional loyalty) on continued intent to donate, and the impact of interaction rituals (participatory interaction, feedback interaction) on emotional energy (emotional attachment, emotional loyalty) in the context of social media. The conclusion of this study is that emotional attachment is positively related to users' continued intent to donate (H1), emotional loyalty is positively related to users' continued intent to donate (H2), feedback interaction is positively related to users' emotional attachment (H3), participatory interaction is positively related to users' emotional attachment (H4), and participatory interaction is positively related to users' emotional loyalty (H6). However, feedback interaction is not positively related to users' emotional loyalty, that is, H5 is not supported. It may be that the user's voluntary participation is more effective than feedback from both parties. Previous studies have also found that consumers have different opinions on whether social media interaction has a positive impact on loyalty [50], indicating that not all interactions can effectively cultivate emotional loyalty.

In addition, price consciousness has been shown to play a negative moderating role between emotional attachment and continued intent to donate (H7). However, price consciousness does not play a negative moderating role between emotional loyalty and continued intent to donate (H8). It may be that users who are emotionally loyal to content creators do not care too much about price, and price

consciousness cannot have a negative moderating effect between emotional loyalty and continued intent to donate.

## 6. Conclusions

In the context of social media, this paper uses attachment theory, customer loyalty theory, and interaction ritual chains as the theoretical model with which to study social media users' continued intent to donate. It was found that emotional attachment and emotional loyalty had significant positive effects on users' continued intent to donate, and participatory interaction had significant positive effects on emotional attachment and emotional loyalty, while feedback interaction had a significant positive effect on emotional attachment. In addition, price consciousness did not have a direct effect but had a negative moderating effect between emotional attachment and continued intent to donate.

### 6.1. Theoretical Implications

Compared with existing studies, the significance of this study is as follows: first, previous studies did not focus on the sustainability of social media users' donations. We studied social media users' continued intent to donate and explained the sustainability of this behavior in the context of social media. This provides new ideas regarding users continuing to donate in the context of social media and enriches research on continued donation. Second, the previous models of social media users' continuous donation used expectation confirmation theory [51], the technology acceptance model, and the theory of planned behavior [52], focusing on user perception and satisfaction [53]. Based on attachment theory, customer loyalty theory, and interaction ritual chains, this study established a new theoretical model for social media users' continued donations, distinguishing the differences between two types of interactions (feedback and participatory interaction) on social media, which was a new application of interaction ritual chains in the field of social media. Third, this paper explained the moderating mechanism of price consciousness, confirming its particularity in the context of social media; that is, it did not have a direct effect, but had a negative moderating effect between emotional attachment and users' continued intent to donate on social media.

### 6.2. Practical Implications

The practical significance of this study is that, in the context of social media, consumers' attention is short and distracted, and content creators should develop and maintain interactive behaviors with users. Social media platforms can maintain the sustainable development of their business model through continued donation. In terms of the type of interaction, participatory interaction is more important than feedback interaction. Content creators and social media platforms should mobilize users' participatory interaction and let them participate in interaction rituals spontaneously, which is conducive to stimulating their continued intent to donate. Second, social media platforms and content creators should recognize the importance of users' emotional responses and use them to meet the emotional demands of their audience, because emotions have an important impact on future purchasing intentions [54]. Finally, social media platforms and content creators should develop different marketing strategies based on user traits.

### 6.3. Limitations and Future Research

The limitations of this study are as follows: first, the samples of this study took WeChat in China as the research object. The results of this study have certain limitations, and external validity is affected to a certain extent. In the future, more diverse samples can be selected to study different cultures and platforms. Second, the variable measurement in this study adopted the form of a self-reporting questionnaire, which might be subject to the influence of common method bias. Future research can utilize multiple sources to obtain data. Finally, this study only explored users' continued intent to donate in the context of social media. There is no further study on the follow-up behavior of continued donation. There is a gap in research between the continued intent to donate and the behavior of

donation. Future research should measure the real donation data of the platform and explore the relationship between users' continued intent to donate and their behavior of continued donation on social media.

**Author Contributions:** Conceptualization, Y.L.; Methodology, L.Y.; Supervision, Y.L.; Writing—original draft, L.Y. All authors have read and agreed to the published version of the manuscript.

**Funding:** This research was supported by the Social Science Fund of Chongqing Federation of Social Science Circles (grant number 2019YBGL065); the Undergraduate Scientific Research Training Program of Chongqing University of Posts and Telecommunications (grant number K2019-107).

**Conflicts of Interest:** The authors declare no conflicts of interest.

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
