# Peer review of "Factors Influencing Social Media Users’ Continued Intent to Donate"

_sustainability, doi:10.3390/su12030879_

Round 1

Reviewer 1 Report

I have now examined the manuscript „Factors Influencing Social Media Users’ Continued Intent to Donate” and I think that the authors carry out an interesting work which presents social media donation. But, the grammatical errors in the paper made it a difficult read. Improper use of prepositions, pronouns, spelling and sentence structure needs correction. In addition, the statistical analysis is not clear presented. Please explain in the text the signify of notations (X1, X2, …Y4, Y5, …) and number (0.84, 0.85, …) from the Figure 2.

In the Abstract, please describe briefly the main methods applied.

In the Methodology section:

Page 6, line 229, 230, 231, 232:  Please write … is defined by … instead of … comes from ….

Page 6, line 255: In the title of Figure 2 correct is … standardized path coefficients … instead of … standardized path …

Page 7, line 265: Please mention why the H5 hypothesis is not supported. Which is the condition that the hypothesis is accepted?

Page 7, line 268: As notes, please added: R2, coefficient of determination and VIF, variance inflation factor

Page 7, line 269: Which are the hypotheses tested? Please mention the hypotheses.

Page 7, line 272: What represent the expansion coefficient of the different variables and value less than 2?

Page 7, line 274, 276, 278: Correct is R2 instead of r2

Please include Conclusions section Because the discussion is complex,.

In the text, reference must be numbered in order, for example [1] instead [25], [2] instead [48] ….

Concerning the References section, the references must be numbered in order of appearance in the text.

The reference [42] Shahin Sharifi, S. Impacts of the trilogy of emotion on future purchase intentions in products of high …. is no cited in the text. Please cited the reference in manuscript or delete it from the References section.

Finally, English needs carefully reviewing.

Reviewer 2 Report

The authors attempted to propose innovative concept and research regarding factors influencing social media users’ continued to intent to donate.

Statistical analysis is irrelevant in the context of an insignificant number of only 466 valid questionnaires. China ranks number 1 in the list of world’s countries by population (almost a billion and a half inhabitants) and the authors discuss about a few hundred of valid questionnaires. The distribution by gender, men vs. women is also very questionable. The limits of this article are significant.

The authors also did not provide sufficient evidence on literature review to support the hypotheses. The Literature review section is practically non-existent being mentioned only a few bibliographic references quite uncorrelated.

The research design is not documented properly and is not justified.

Therefore, it is specifically recommended to:

take into consideration more current publications in the sphere of discussed subject matter, focus considerations conducted in the paper more on the category of sustainable development, develop the discussion of results and the conclusions resulting from conducted research, deepen the description of the limitations of conducted research and indicate the trends for further empirical research, expand the managerial implications in the article.

Reviewer 3 Report

The presented paper is well written, solid and valuable analysis of social media users’ continued intent to donate phemomena. In my opinion, there is a need in conducting research in this area and undertaking attempts to explain the sustainability of social media users’ donations.
   Authors proposed a new theoretical framework by integrating attachment theory, customer loyalty theory, and interaction ritual chains. The structural equation modelling (SEM) were used to construct and verify the cause-effect model between the selected variables: Feedback interaction, Emotional attachment, Participatory interaction, Emotional Loyalty, and Continued intent to donate. It resulted in the development of an authors’ empirical model with 6 detailed hypotheses.
   Among the weaknesses of the presented text - in my opinion - should be indicated:
- No explanation regarding the chosen SEM method of data analysis. It is worth stressed that structural equation modelling method allowed verifying the theoretical hypotheses put forward based on the correlations between particular variables, both in their occurrence and also their strength and direction.
- Authors should name their model proposal, e.g., Model of users' continued intent to donate. (Figure 1. p. 2, line 76).
- "SEM" can be added in keywords and in the abstract, the method of data analysis used can be emphasized.

- Conclusions section should be expanded about suggestions for the charity sector from the conducted research?

   In my opinion, the article would be suitable for publication, however, after completing the above information and answers to the above detailed questions.

   Thank you for the opportunity of reading and reviewing very interesting article.

Round 2

Reviewer 1 Report

However, I recommend some minor correction before it publication:

Page 6, line 314-322: Please write p instead of P.

Reviewer 2 Report

The article has been significantly improved from the original version.
